# SKATR: A Self-Supervised Summary Transformer for SKA

Ayodele Ore[1], Caroline Heneka[1], and Tilman Plehn[1,2]

**1** Institut für Theoretische Physik, Universität Heidelberg, Germany
**2** Interdisciplinary Center for Scientific Computing (IWR), Universität Heidelberg, Germany

March 17, 2025

## Abstract

The Square Kilometer Array will initiate a new era of radio astronomy by allowing 3D imaging of the Universe during Cosmic Dawn and Reionization. Modern machine learning is crucial to analyze the highly structured and complex signal. However, accurate training data is expensive to simulate, and supervised learning may not generalize. We introduce a self-supervised vision transformer, SKATR, whose learned encoding can be cheaply adapted for downstream tasks on 21cm maps. Focusing on regression and generative inference of astrophysical and cosmological parameters, we demonstrate that SKATR representations are maximally informative and that SKATR generalizes out-of-domain to differently-simulated, noised, and higher-resolution datasets.

# 1  Introduction

The Square Kilometer Array (SKA) is a discovery machine that has recently seen first light. The SKA-LOW part of the radio interferometer is sensitive to the 21cm background fluctuations of neutral hydrogen during the Epoch of Reionization (EoR) and Cosmic Dawn (CD). Its large-scale signal is sensitive to the thermal and cosmological evolution of the Universe, making it a tracer also of primordial sources of radiation and fundamental physics [1–9].

The SKA will provide deep 3D-imaging of over 50% of the observable Universe. This tomography of the large-scale structure will produce data rates of several TB/s and archival data of up to hundreds of PB/a [10]. At the same time, the SKA data suffers from a complexity problem: foregrounds and systematics from interferometric reconstruction cannot be modeled accurately for synthetic data; simulations via hydrodynamical and radiative transfer or approximate hydrodynamical (semi-numerical) simulations suffer from model uncertainties [11–16]; and an analytical optimal summary to construct a likelihood does not exist.

Classical analyses use physics-motivated highly compressed summaries. Analyses based on the power spectrum assume Gaussianity, motivated by CMB assuming standard cosmology. This assumption breaks down for the highly non-Gaussian SKA signal. Beyond-Gaussian statistics such as bispectra and morphological diagnostics improve constraints and reduce bias for parameter inference from 21cm intensity maps [17, 18]; a picture which might revert when faced with foregrounds, as these methods do not generalize [19].

Modern machine learning opens a path toward optimality in data-intensive analyses in fundamental physics and cosmology [20–26], including optimal compression for robust performance at downstream tasks such as inference [27]. Especially for SKA data we need to bridge different simulators and assumptions on noise and systematics, while remaining maximally informative [28]. Supervised approaches may not generalize well enough, and the size of realistic simulated datasets is limited.

We propose to use self-supervised learning to train a maximally-informative summary network that can easily be adapted for downstream tasks without fine-tuning. Self-supervised approaches based on contrastive learning or masking have been shown to generate expressive representations, both in vision [29–37] and fundamental physics [38–41]. Similarly, self-supervised learning has been shown to aid data compression and inference for galaxy surveys [42, 43], including the use of pre-trained foundation models [44]. Given the large volume probed, systematics from radio interferometric measurements, and the less known adequate summary statistics for 21cm physics, we expect the benefits to be even larger for SKA data.

In this analysis, we focus on regression and inference of a set of astrophysical and cosmological parameters from SKA lightcones. We first establish that Vision Transformers [45] are a suitable network architecture by comparing with the current CNN benchmark [46, 47]. We then show that our self-supervised SKA Transformer (SKATR) learns a near-lossless compression. In particular, a shallow MLP trained on frozen SKATR summaries matches the performance of a ViT trained from scratch, at much higher efficiency. Using datasets simulated at different resolutions, we find that SKATR generalizes well when faced with parameter information absent during the pre-training, as well as with instrumental and thermal noise and foreground avoidance. Moreover, SKATR generalizes better than a summary pre-trained with full supervision.

The remainder of the paper is organized as follows. Section 2 details our simulated datasets as well as the transformations we employ for preprocessing and data augmentation. In Section 3 we introduce the network architecture and self-supervised training strategy that comprise SKATR. A series of results showing the benefit of SKATR are presented in Section 4. In par-

ticular, we show that the fixed SKATR summary a) matches supervised benchmark networks in regression and inference tasks (Section 4.2 and Section 4.3), b) generalizes out-of-domain regarding noise and new parameter correlations present in higher resolution data (Section 4.4), c) outperforms supervised baselines when data is limited (Section 4.5), d) performs well when resolution adaptation is implemented (Section 4.6). Finally, we give concluding remarks in Section 5.

## 2 Lightcones

We work with two lightcone (LC) datasets simulated with the semi-numerical code 21cm-FASTv3 [48]. An LC is a discrete 3-dimensional field of 21cm brightness offset temperature fluctuations $\delta T_b(\mathbf{x}, \nu)$ over on-sky coordinates $\mathbf{x}$ and frequency $\nu$. For our analysis, we focus on six model and simulation parameters, two determining the cosmology, two sensitive to EoR astrophysics [46], and two to cosmic dawn astrophysics:

- $m_{\mathrm{WDM}} \in [0.3, 10]\,\mathrm{keV}$: the lower limit on the warm dark matter mass allows for a small tension with cold dark matter (CDM), current astrophysical constraints point towards the upper limit [49, 50]. Here, structure formation looks similar to CDM, because the free-streaming length is inversely proportional to $m_{\mathrm{WDM}}$;

- $\Omega_{\mathrm{m}} \in [0.2, 0.4]$: the dark matter density parameter controls structure formation. The range for training is deliberately chosen wider than Planck limits [51] but encloses them;

- $E_0 \in [100, 1500]\,\mathrm{eV}$: the X-ray energy threshold for self-absorption by host galaxies, where X-rays with energies below $E_0$ do not escape the host galaxy;

- $L_{\mathrm{X}} \in [10^{38}, 10^{42}]\,\mathrm{erg\,s^{-1}\,M_\odot^{-1}\,yr}$: the specific integrated X-ray luminosity $< 2\,\mathrm{keV}$ per unit star formation rate that escapes host galaxies;

- $T_{\mathrm{vir}} \in [10^4, 10^{5.3}]\,\mathrm{K}$: the minimum virial temperature (related to a minimal virial halo mass) needed for cooling within halos to enable star formation;

- $\zeta \in [10, 250]$: the ionization efficiency, given by

$$\zeta = 30 \frac{f_{\mathrm{esc}}}{0.3} \frac{f_\star}{0.05} \frac{N_{\gamma/b}}{4000} \frac{2}{1 + n_{\mathrm{rec}}} \,, \tag{1}$$

in terms of the escape fraction of ionizing photons into the intergalactic medium $f_{\mathrm{esc}}$, the fraction of galactic gas in stars $f_\star$, the number of ionizing photons per baryon in stars $N_{\gamma/b}$, and the typical number density of recombinations for hydrogen in the intergalactic medium $n_{\mathrm{rec}}$.

Our simulations sample parameters points from flat priors in the ranges given above. For all other cosmological parameters we refer to the Planck measurements, assuming flatness and a cosmological constant. The central values are $\Omega_{\mathrm{b}} = 0.04897$, $\sigma_8 = 0.8102$, $h = 0.6766$, and $n_s = 0.9665$ [52].

### 2.1 Datasets

**High-resolution (HR)** The first of our datasets [27, 46] consists of 5k LCs with spatial size $200 \times 200\,\mathrm{Mpc}^2$ and redshift range $z \in [5, 35]$. It is simulated at a spatial resolution of $1.42\,\mathrm{Mpc}$, leading to LCs with 140 voxels along each on-sky axis and variable length (depending on $\Omega_{\mathrm{m}}$) in the redshift axis. To standardize the LC shapes, we keep the first 2350

| Dataset | HR | HRDS | LR |
|---|---|---|---|
| LC Shape | $(140, 140, 2350)$ | $(28, 28, 470)$ | $(28, 28, 470)$ |
| Simulated resolution [Mpc] | 1.42 | 1.42 | 2.84 |
| Downsample factor | - | 5 | 2.5 |
| Noised version available | ✓ | ✓ | ✗ |
| Filter valid LCs | ✓ | ✓ | ✗ |
| Total LCs | 5k | 5k | 35k |

Table 1: Details of the high-resolution (HR), HR-downsampled (HRDS), and low-resolution (LR) lightcone datasets.

voxels. Consequently, only LCs with $\Omega_{\mathrm{m}} = 0.4$ entirely span $z \in [5, 35]$, while the rest end at $z < 35$. Since the prior ranges given above are conservative, a small fraction of LCs display unrealistic reionization histories. For example, some parameter combinations result in LCs with a Thomson scattering optical depth inconsistent with Planck [52] at more than $5\sigma$, or late reionization such that the mean intergalactic-medium neutral fraction $\bar{x}_{\mathrm{HI}}$ is over 0.1 at redshift $z \sim 5$, in strong tension with Ly$\alpha$ forest observations [53]. Our 5k LCs are filtered to satisfy these criteria.

**Noised LCs**    Mock observed noised realizations of the 5k HR LCs are created by splitting each LC into smaller coeval boxes than were used during simulation. Each coeval box corresponds to a fixed redshift. At each redshift the expected noise power is estimated using 21cmSense [54, 55]. The thermal and instrumental noise estimate is based on ~1000 hrs of SKA-Low stage 1 tracked observations, and the noise power is added to the Fourier-transformed coeval boxes. We follow a foreground avoidance strategy where galactic and extragalactic foregrounds to the 21cm signal are localized in $k$-space in the so-called 21cm foreground wedge [56, 57]; we assume the wedge covers the primary field-of-view of the instrument. These modes are zeroed out before transforming the full boxes of signal plus noise to real space to obtain noised LCs.

**Low-resolution (LR)**    A further 35k LCs were simulated using identical prior ranges for the parameters, but with a coarser resolution of 2.84 Mpc. This yields lightcones with shape $(70, 70, 1175)$. To alleviate computational demand, we then downsample these images by a factor of 2.5 in each axis using `transforms.resize` from the `skimage` package. This results in lightcones with shape $(28, 28, 470)$. Unlike the HR datasets, we do not make any selection based on the validity of the reionization history.

**HR-downsampled (HRDS)**    Finally, we construct a downsampled version of the HR dataset with image size $(28, 28, 470)$ to match the LR dataset. This is achieved by averaging over $5 \times 5 \times 5$ groups of voxels. This HRDS dataset is superior to the LR dataset, since the information available to the forward simulation degrades with lower simulated resolution. As we will see later, the main difference is that the LR dataset has no sensitivity to $m_{\mathrm{WDM}}$ simply because of its reduced simulation resolution. This parameter, as well as $T_{\mathrm{vir}}$, place a threshold on early star formation, via the Jeans mass for $m_{\mathrm{WDM}}$, and are thus degenerate. Because the LR dataset does not include information on $m_{\mathrm{WDM}}$, the second parameter $T_{\mathrm{vir}}$ can be extracted perfectly without this parameter degeneracy present. Once limited knowledge about $m_{\mathrm{WDM}}$ is introduced into the HRDS dataset, the extraction of $T_{\mathrm{vir}}$ becomes much less precise and prone to outliers. For further discussion, see Appendix A.

A summary of the three datasets is given in Table 1. In the LR dataset we reserve 1k LCs for testing and 25k LCs for training, with the remainder used as validation. For the HR and

HRDS datasets, the testing splits consist of 750 LCs and up to 3.75k are used for training.

## 2.2   Preprocessing and augmentations

We perform simple shift and scale transformations to preprocess LC voxel values and parameter labels into the range $[0, 1]$,

$$x \rightarrow \frac{x - x_{\min}}{x_{\max} - x_{\min}} \ . \tag{2}$$

For the simulation parameters, the minimum and maximum values correspond to the prior boundaries. For the voxels we use $(x_{\min}, x_{\max}) = (-120, 1)$ universally.

To boost the statistical power of the datasets, we employ data augmentation based on symmetries of the LCs. At each training iteration we randomly sample a transformation by composing 90° rotations around the redshift axis with an optional reflection in the spatial axes. The identity is included in the set of possible transformations. These augmentations are beneficial for every task discussed below.

## 3   SKATR

### 3.1   Vision transformer

Transformers are a powerful network architecture for processing sequence data [58] and have proven useful in fundamental LHC physics [22, 59]. They can be adapted to non-sequence data by using specialized positional encodings, which are necessary to break the permutation-equivariance of the attention mechanism. For example, constructing encodings based on a grid in two or more dimensions allows application to images, leading to the Vision Transformer (ViT) [45, 60].

Figure 1 shows how a ViT processes images. First, an image is divided into non-overlapping patches of pixels. Each patch is embedded into a high-dimensional space using a shared linear projection, then augmented with an encoding of the patch location in the image. The set of patch representations are processed by an alternating sequence of multi-head attention and feed-forward operations. Normalization layers and skip connections are also used to stabilize optimization.

In a ViT, the patch size controls a trade-off between model expressivity and complexity. While using smaller patches probes spatial correlations in the input image at finer scales, this also leads to a larger number of elements entering the attention operation. In order to manage the computational cost of a ViT, the patch size should be selected with an expected image

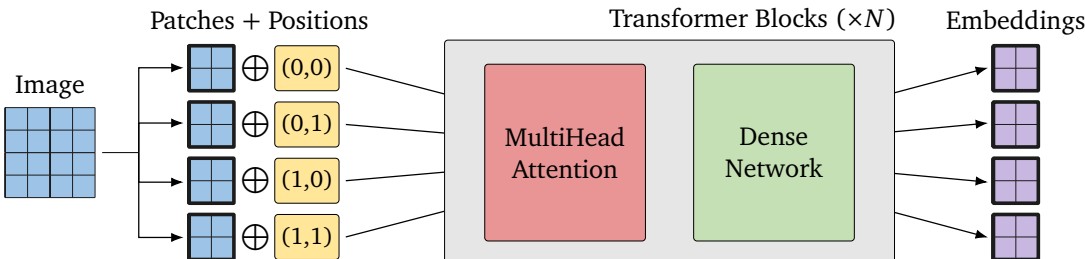

Figure 1: Schematic diagram of a vision transformer encoder using $2 \times 2$ patches.

resolution in mind. In our analysis, we use patch sizes of (4,4,10) for downsampled LCs, and (7,7,50) for full-resolution LCs in the HR dataset. We arrive at these values by hand; they approximately maximize the total number of patches (with a similar number in each axis) while remaining within memory limits.

Depending on the task, a loss can be calculated directly using the set of patch embeddings output by the ViT. This is the case for our pretraining, to be presented in the following section, as well as other common tasks such as segmentation or diffusion. For cases where a global feature vector is needed, such as in regression, an aggregation step can be used to obtain a single input for the task-specific head network. There are a number of possibilities for this aggregation. A simple option is a mean over patch embeddings $z$ followed by a two-layer dense network,

$$\text{MLP}(z) \equiv W_2\,\text{ReLU}(W_1\bar{z})\,, \tag{3}$$

where $\bar{z}$ is the average of $z$ over patches, and $W_1$ and $W_2$ are weight matrices with respective shapes $d \times d$ and $6 \times d$ for transformer embedding dimension $d$. We use this setting for all regression results in our analysis, where the output dimension 6 corresponds to the number of target parameters. Another, more flexible, possibility is to learn a dynamic pooling function using a cross attention layer,

$$\text{XAttn}(z) \equiv \text{Softmax}\left(\frac{q^T W_K z}{\sqrt{d}}\right) W_V z\,, \tag{4}$$

with $d \times d$ key and value weight matrices $W_K$ and $W_V$, and a learnable $d \times 1$ input token $q$. For the results in this work, we find the simple mean and MLP option to be sufficient.

## 3.2 Self-supervised pre-training

Lightcones are represented in a high-dimensional voxel space, which we expect to be compressible. For any kind of analysis, we need encoders to map the voxels to a lower-dimensional embedding space. The goal of self-supervised pre-training is to learn an encoding $f_\theta(x)$ that produces informative representations of the data $x$ without using any labels or, in our case, model parameters.

To train such an encoder, we adopt a self-supervised learning framework based on Joint Embedding Predictive Architectures (JEPA) [36, 37]. Our SKA Transformer (SKATR) setup is shown in Figure 2 and involves two ViTs: a context encoder $f_\theta$ and a target encoder $g_\varphi$. The networks share identical architectures and are initialized with the same weights. During training, an LC (batch) is divided into a set of $n$ patches $x = \{x_i\}_{i=1}^n$. A masked view $\tilde{x}$ is generated by dropping a sampled set of patch locations $M$

$$\tilde{x} = \{x_i \in x \,|\, i \notin M\} \subset x\,. \tag{5}$$

The masked and original LCs are embedded with the context and target networks respectively,

$$\tilde{z} = f_\theta(\tilde{x}) \qquad \text{and} \qquad z = g_\varphi(x)\,. \tag{6}$$

Finally, a transformer $h_\psi$ with smaller hidden dimension than the encoders predicts the target patch embeddings given the context embeddings,

$$p = h_\psi(\tilde{z})\,. \tag{7}$$

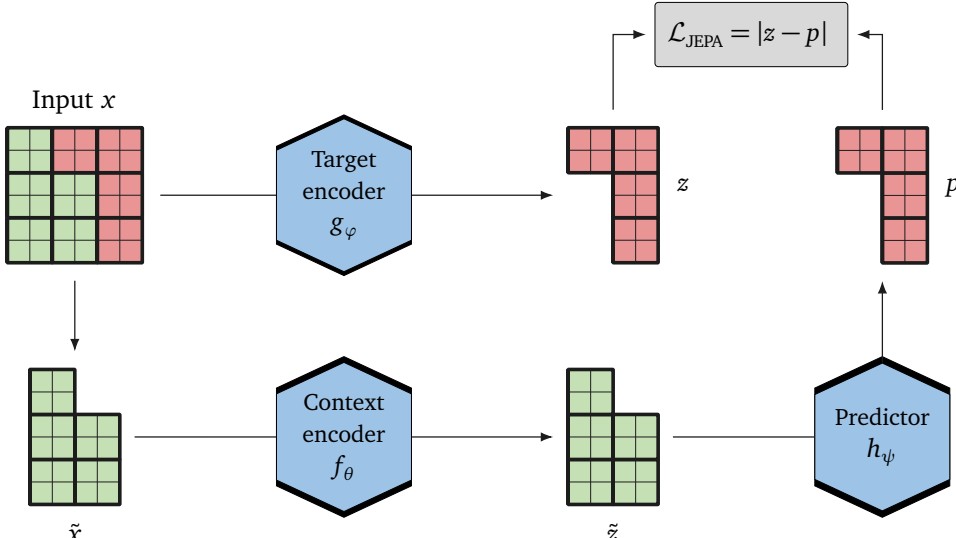

Figure 2: Illustration of the pre-training. At each training iteration, a mask is sampled that defines context (green) and target (red) patches. After training, the context encoder is taken as the summary network.

The loss is the mean absolute error between $p$ and $z$ at the locations of the masked patches,

$$\mathcal{L}_{\text{JEPA}} = \left\langle \frac{1}{|M|} \sum_{i \in M} \left| g_\varphi^i(x) - h_\psi^i(f_\theta(\tilde{x})) \right| \right\rangle_{p_{\text{data}}(x), p_{\text{mask}}(M)}, \tag{8}$$

where $p_{\text{mask}}(M)$ encodes a user-defined masking strategy (see Appendix B for details). Given that the above loss is not contrastive, there is a risk of representation collapse if we optimize it with respect to all parameters $\theta$, $\varphi$, and $\psi$. To avoid this, only the context encoder and predictor parameters are updated via gradient descent. The target encoder parameters instead follow the exponential moving average of the context encoder parameters,

$$\varphi_{i+1} = \tau \varphi_i + (1 - \tau) \theta_i, \tag{9}$$

where $\tau$ is a momentum hyperparameter controlling the rate at which the target encoder parameters are updated.

Once trained, a global summary of an LC can be constructed by passing it through the context encoder without a mask, then taking the mean over the resulting patch embeddings. Throughout our analysis we always pre-train SKATR on the LR dataset, with an embedding dimension of 360. This leads to a highly-compressed SKATR representation in $\sim 10^3$ times fewer dimensions than the original LC. We summarize the proposed training pipeline in Figure 3. For the complete set of hyperparameters, see Appendix B.

## 4 Results

To understand and quantify the behavior of SKATR as an SKA summary network we use regression and inference of six cosmological and astrophysical parameters, described in some detail in Section 2,

$$y \equiv \{ m_{\text{WDM}}, \Omega_{\text{m}}, E_0, L_{\text{X}}, T_{\text{vir}}, \zeta \}. \tag{10}$$

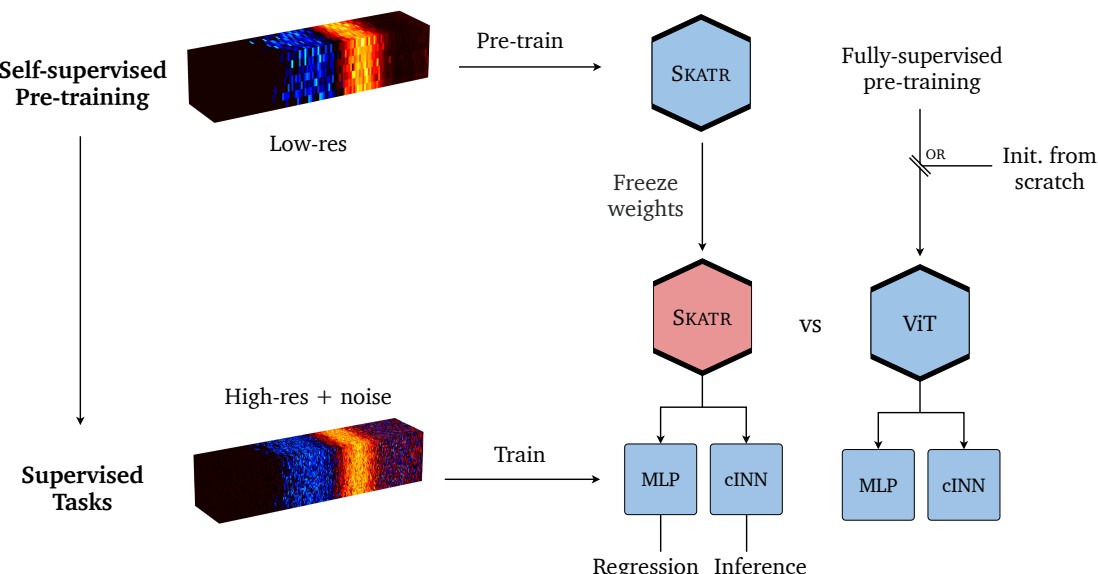

Figure 3: Summary of the pipeline for our self-supervised SKA Transformer (SKATR). Red shading indicates that the SKATR backbone is frozen for supervised tasks.

The benchmark for this regression task is given by the 3D-21cmPIE-Net, a CNN [46, 47]. This network is also used as a summary network for the corresponding generative inference [27], where it serves the same function with respect to data compression as a foundation model. In all cases below, the SKATR network is pre-trained as described in Section 3, then frozen.

## 4.1 ViT performance

First, we demonstrate the power of the ViT by comparing with the established CNN [46, 47] for the HR dataset. The CNN consists of a series of 3D-convolutional blocks that gradually downsample input LCs, followed by global mean and a 4-layer MLP. We train both networks to regress our six simulation parameters using the normalized mean absolute error (NMAE) loss,

$$\text{NMAE} = \left| \frac{y_{\text{pred}} - y_{\text{true}}}{y_{\text{max}} - y_{\text{min}}} \right|. \tag{11}$$

We also measure performance in terms of the NMAE. This allows us to compare errors across different parameters, irrespective of their magnitude or simulated prior range.

The regression results using the HR dataset are shown in Figure 4, where the predictions of both networks are shown against the true value for the simulated lightcone in each parameter. To improve visual clarity, points are binned by the true parameter label and the CNN and ViT predictions are slightly offset from one another horizontally. The lower subpanels show the NMAE following the same binning (without error bars), and the horizontal dashed line indicates the mean error over all test points.

The ViT and the CNN benchmark both regress $\Omega_{\text{m}}$, $L_{\text{X}}$ and $\zeta$ well, while the remaining parameters are more difficult. In particular, the LCs lose sensitivity to $m_{\text{WDM}}$ and $E_0$ above thresholds of 3 keV and 1 keV respectively. At these points the network predictions plateau. $T_{\text{vir}}$ is regressed poorly due to its degeneracy with $m_{\text{WDM}}$, as discussed in Appendix A. Comparing the two networks, we see that, the ViT indeed extracts information from the LCs at a level at least as strong as the 21cmPIE-Net.

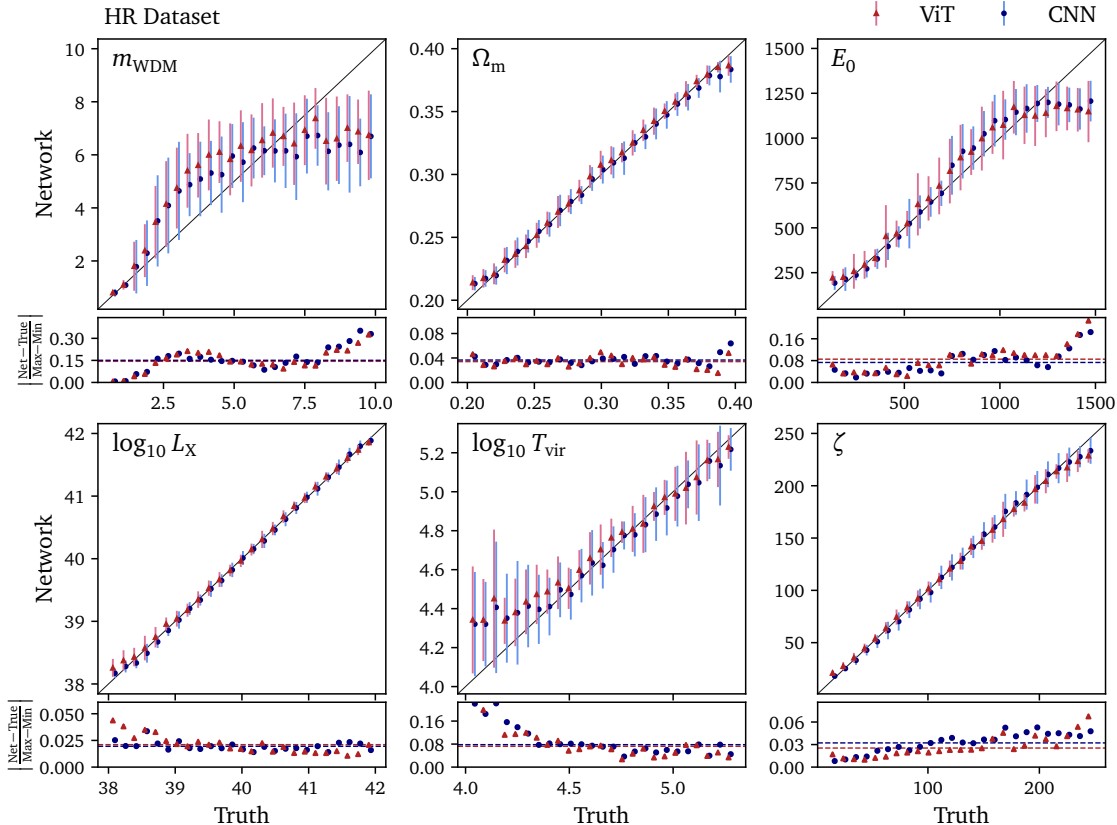

Figure 4: Performance for our **ViT (red) vs the CNN benchmark (blue)**, both trained from scratch to regress simulation parameters on the HR dataset. Network predictions on the test set are binned by the true parameter value, and points show the mean $\pm 1\sigma$ in each bin. The sub-panel shows the mean absolute error per bin normalized to the simulated parameter ranges.

## 4.2 SKATR regression

Now we examine the performance of the LC summary learned by SKATR. We again look at parameter regression, but this time comparing a ViT (trained from scratch) to a lightweight network trained on SKATR-summarized LCs using the LR dataset. Since the SKATR backbone will not be trained in this stage, we evaluate the summary once on each LC and save the resulting dataset. This dataset is then used to train the small network, whose architecture we match with the 2-layer MLP from Eq. (3). To implement data augmentation in this scheme, we also summarize all transformations of a given lightcone (see Section 2.2). The data loading is then customized to select a random transformation in each batch during training.

The results for the LR dataset are shown in Figure 5. Due to the coarse resolution, $m_{\mathrm{WDM}}$ is no longer predictable at any point in the prior range. This resolves the correlations and degeneracies present in the HR dataset and allows both networks to regress the $\Omega_{\mathrm{m}}$, $L_{\mathrm{X}}$, $T_{\mathrm{vir}}$, and $\zeta$ parameters extremely precisely and slightly improve $E_0$.

Comparing the ViT, trained from scratch, with the MLP acting on SKATR-summarized LCs, the compressed representation achieves equal or smaller average error. This demonstrates that the SKATR summary retains all relevant information. Further, because the MLP has so few parameters, its training is more stable and converges much faster than the ViT. In Figure 6 we illustrate this acceleration using the validation loss. For training alone, SKATR leads to a speed

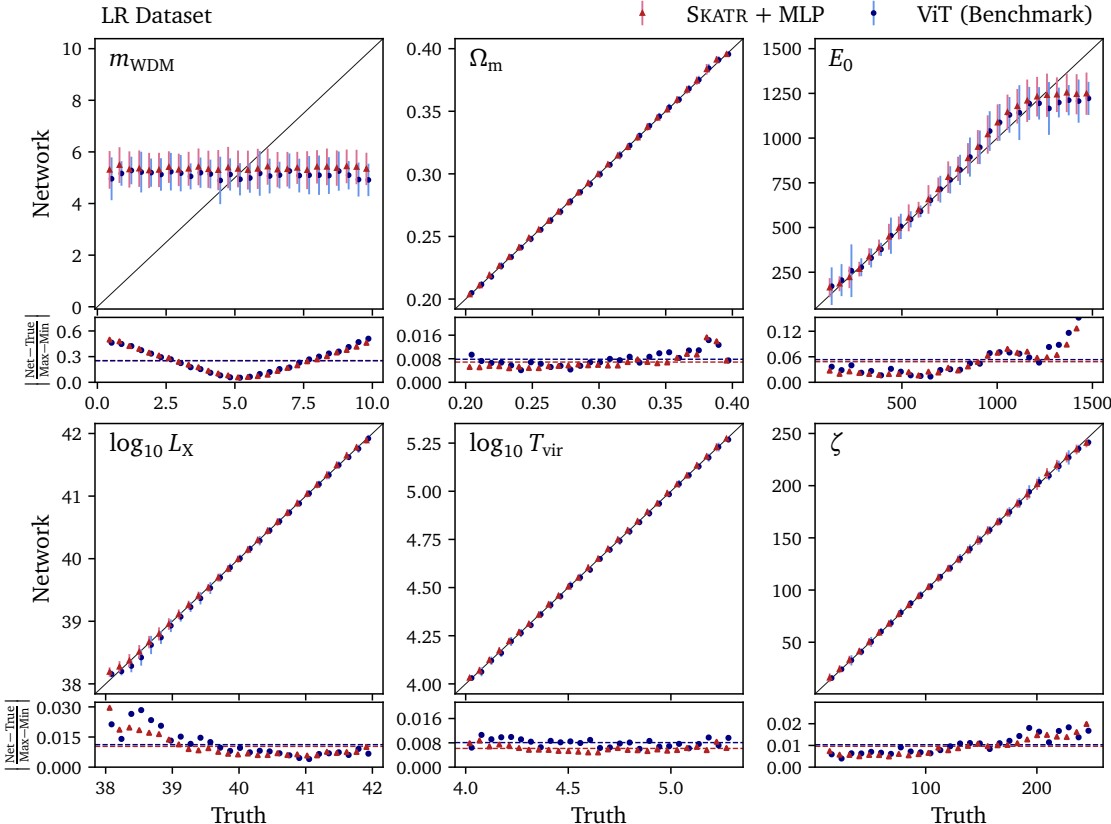

Figure 5: Performance for **frozen SKATR summaries (red) vs ViT benchmark trained from scratch (blue)**, where SKATR is complemented with a 2-layer MLP. All training and testing is performed on the LR dataset.

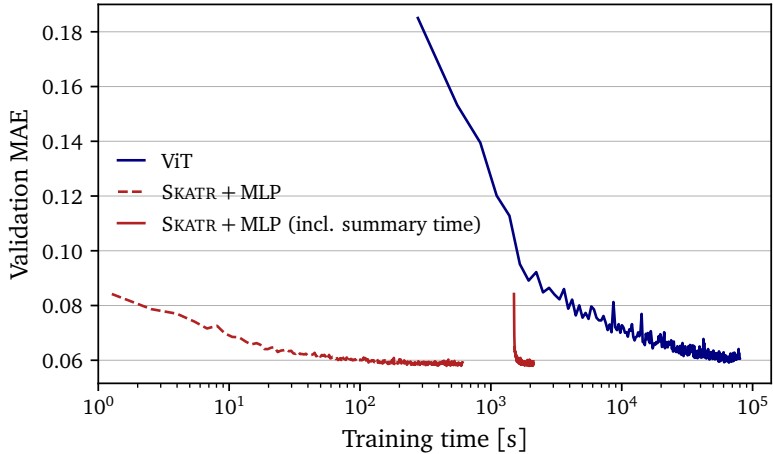

Figure 6: Loss over time for an **MLP with frozen SKATR summaries (red) vs ViT trained from scratch (blue)**, as shown in Figure 5. The solid SKATR line includes the time to summarize the dataset, while the dashed line does not. Each line begins at the end of the first epoch.

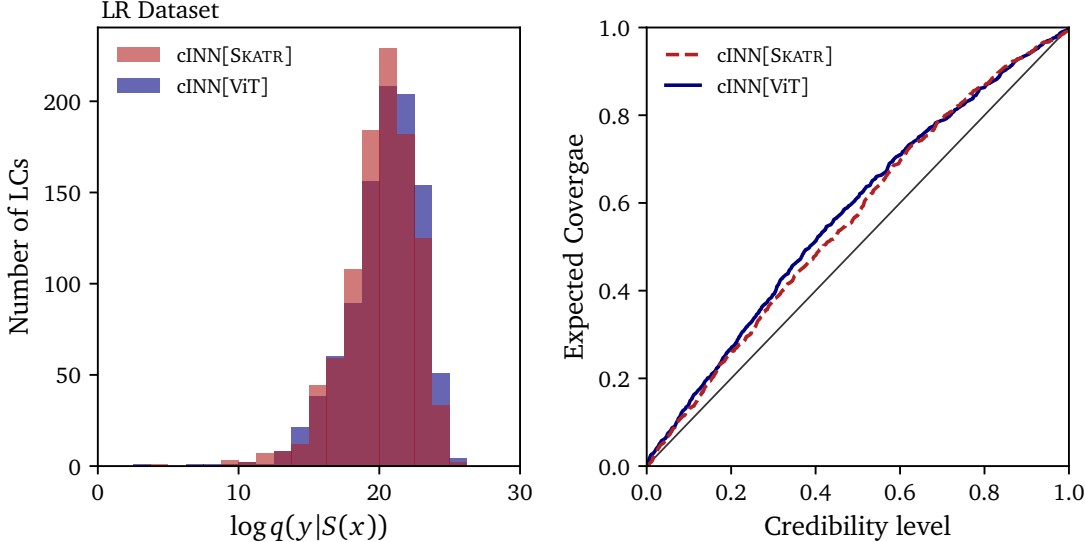

Figure 7: Inference performance for **cINN with frozen SKATR summary vs jointly-trained ViT** on the LR dataset. Shown are the distribution of log-likelihoods (left) and calibration (right) in the test set.

enhancement by a factor of several hundreds, to reach the same performance as the ViT. Even including the time to summarize the dataset, SKATR is roughly a factor 50 faster in training to the final converged performance.

## 4.3 SKATR inference

Next we test SKATR on a more challenging task — inference of the full 6-dimensional posterior distribution $p(y|x)$ of parameters $y$ given data $x$. As in Ref. [27], we train conditional invertible neural networks (cINNs) to approximate the posterior. The conditioning on LCs is always via a summary and so the loss is

$$\mathcal{L}_{\text{cINN}} = -\left\langle \log q_\vartheta\big(y|S_\phi(x)\big)\right\rangle_{p_{\text{data}}(x,y)}, \tag{12}$$

where $q_\vartheta$ is the probability density defined by the cINN, and $S$ is a summary which may be fixed or trainable, with parameters $\phi$.

For $S$, we consider two options. First, a ViT initialized from scratch with no head network. In that case, the summary is the mean of learned patch embeddings. Second, we use the mean embedding from a frozen pre-trained SKATR network. Again, no MLP head network is used and the cINN is conditioned directly on the frozen summary.

To evaluate the constraining power and calibration of the networks, we show the posterior likelihoods and coverage over the LR test set in Figure 7. The distributions of likelihoods (left panel) are closely matched, though a narrow advantage is evident when training from scratch with a ViT. The calibration panel (right) shows the cumulative distribution function of the rank statistic, defined as the fraction of posterior samples with model likelihood larger than the true label. In these terms, an under(over)-confident posterior lies above (below) the diagonal. We see that the posteriors defined by each network are equally conservative, i.e. slightly less precise parameter estimates than possible. In Figure 8 we show the parameter-wise posterior predictions for a sample of LR test points, comparing the cINN with a ViT or the frozen SKATR summary. Both networks perform comparably to the previous results for

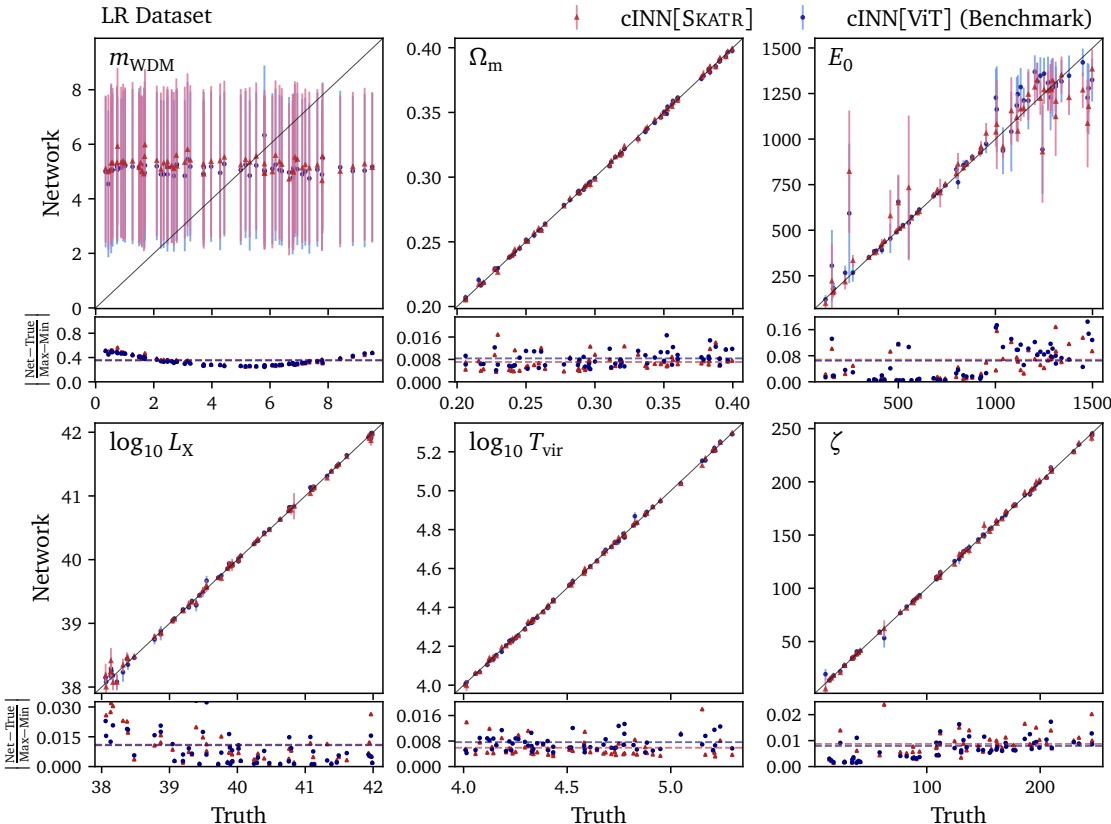

Figure 8: Posteriors using **a frozen SKATR summary vs the jointly-trained ViT benchmark**, acting as summaries for a cINN trained and tested on the LR dataset. For a given test LC, each point shows the posterior mean with $1\sigma$ error bars over 5k network samples.

regression. On average, the SKATR summary gives posterior means that are closer to the truth value in each parameter than the ViT, meaning SKATR performs at higher accuracy.

## 4.4  Generalization

Next, we demonstrate that the performance of SKATR also extends to new datasets. In particular, we are interested in whether SKATR provides a benefit on LCs simulated at higher resolution than the pre-training set, and including those with a noise model. That would allow us to transfer information from cheap simulations to expensive simulations. To this end, we repeat our regression test with the noised HRDS dataset. Here there main challenge and interest will be whether the shallow MLP can regress the $m_{\mathrm{WDM}}$ parameter given the SKATR summary, as $m_{\mathrm{WDM}}$ cannot be predicted from the LR dataset used to train SKATR due to insufficient resolution.

In Figure 9 we compare the SKATR-MLP combination to a ViT trained from scratch on the noised HRDS dataset. Once again, we see that the SKATR summary matches the regression performance of a full ViT training. A slight advantage for the MLP is apparent in $\Omega_{\mathrm{m}}$, likely due to the large dataset used for SKATR pre-training. Since SKATR displays no obvious failures, we conclude that the fixed summary is sufficiently general to capture new effects in the LCs due to noise and new correlations in the parameters due to simulation resolution. In particular, the SKATR-MLP combination has no trouble regressing $m_{\mathrm{WDM}}$, demonstrating that SKATR remains informative outside the training domain.

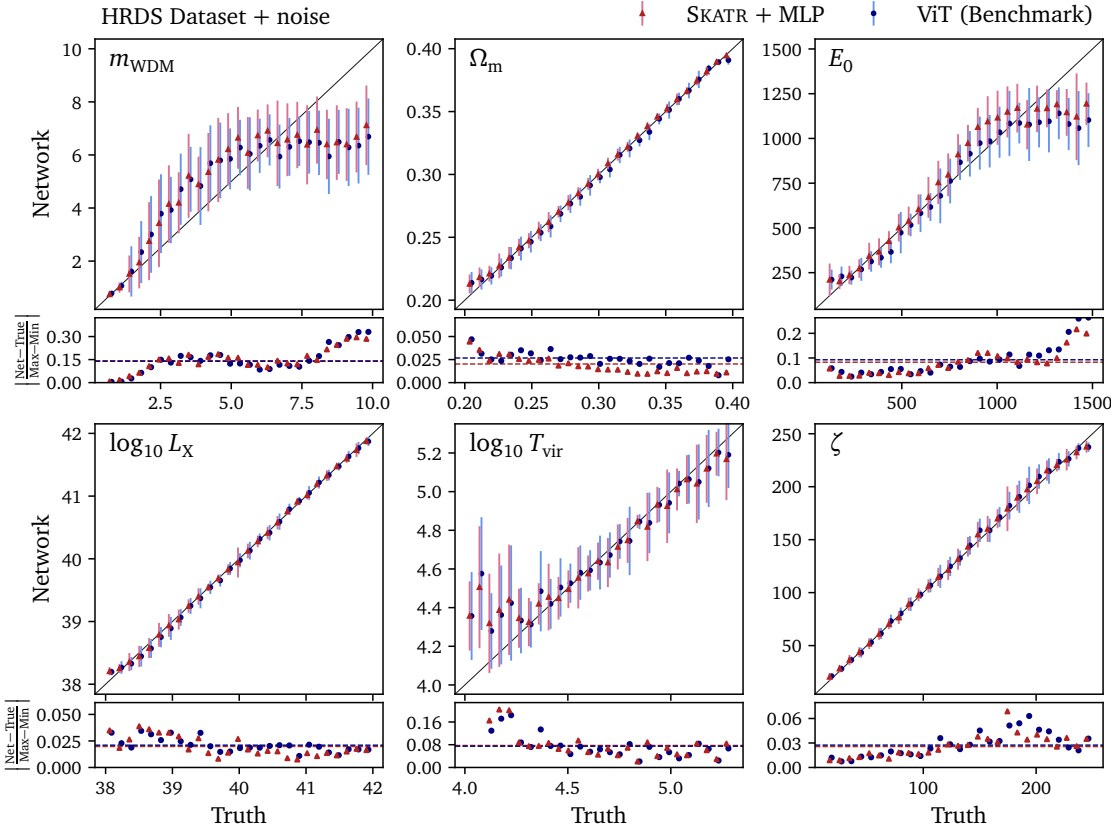

Figure 9: Performance for **frozen SKATR summary vs ViT benchmark trained from scratch** on the noised HRDS dataset, SKATR is complemented with a trained 2-layer MLP. The SKATR pre-training on LR dataset does not contain any information on $m_{\mathrm{WDM}}$.

## 4.5 Data efficiency

As mentioned above, one of the main reasons for pre-training SKATR is that it is extremely efficient when it comes to training for the downstream task, in our case the regression of the model parameters from some test dataset. To illustrate that gain, we emulate data-limited scenarios by scanning a range of training split sizes within the HRDS dataset. In each case, we train a ViT from scratch and compare its performance to a shallow MLP trained on SKATR-compressed LCs. Figure 10 shows that the SKATR summary consistently yields smaller error than the ViT trained from scratch. This is despite being pre-trained out of domain, on LR LCs. The most impressive improvement can be seen for little training data, where the ViT struggles to capture the relevant information. The only exception is in $L_{\mathrm{X}}$, where SKATR is outperformed on the largest training set. The improvement from SKATR over the from-scratch baseline is smallest for $T_{\mathrm{vir}}$, possibly due to the degeneracy with $m_{\mathrm{WDM}}$ introduced in the HRDS dataset. This kind of improvement can be important for SKA, because generating HRDS training data is expensive and will eventually limit the actual data analysis.

To see the specific impact of the self-supervised JEPA training we introduce a second baseline, pre-training a ViT with fully-supervised regression on the LR dataset. From this ViT we drop the regression head and take the mean over patch embeddings as a summary. Also in Figure 10 we see that for most parameters SKATR is significantly better than the supervised backbone. The only exceptions are $L_{\mathrm{X}}$, where the improvement is only marginal, and $E_0$,

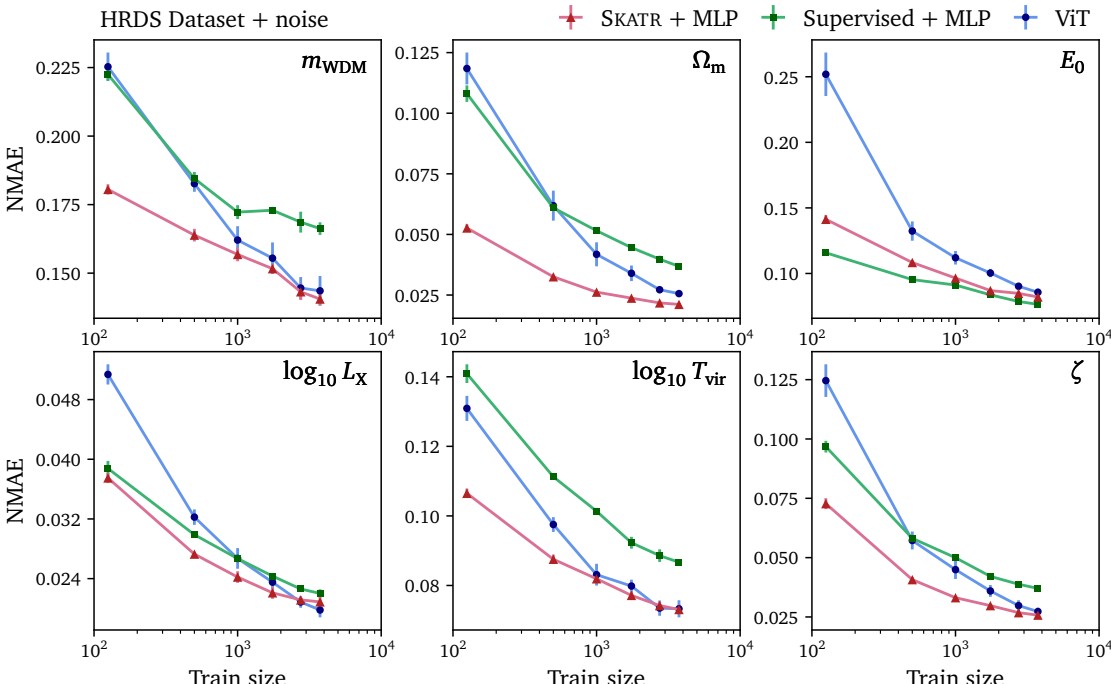

Figure 10: Data scaling for **SKATR-summarized LCs vs supervised-summarized LCs vs a ViT trained from scratch** using the noised HRDS dataset. Shown is the mean regression error as a function of the number of LCs used for training. Each point is the average over ten training runs, with $1\sigma$ error bars.

where SKATR is slightly outperformed. Moreover, the fully-supervised summary network is typically worse than the ViT trained from scratch. While in $\Omega_m$, $E_0$, $L_X$ and $\zeta$ the MLP does eventually achieve a lower error in the smallest training size, this is not the case for $m_{WDM}$ or $T_{vir}$. Recalling that these parameters exhibit the greatest change in behavior between the LR and HR datasets, this observation highlights the difficulty for supervised pre-training. Instead of providing a generalizing summary, it encodes details of the correlations relevant for the specific fully-supervised task. When these correlations are different for the test dataset, the regression-pre-trained model does not generalize. We have also checked that the same results hold for the pure HRDS dataset without noise.

## 4.6 Resolution adaptation

So far, we have shown that a SKATR summary pre-trained on the LR dataset generalizes to new correlations in the HRDS dataset which has been downsampled to match the LR resolution. What remains to be seen is whether the LR-trained SKATR can perform on the full-resolution HR dataset. Due to the factor $5^3$ increase in the number of voxels between these datasets, adapting the resolution comes with computational cost, one way or another. There are a number of options to tackle this problem, which we now discuss in turn.

The most straightforward approach is to split the HR LCs using the same patch shape as for pre-training. This requires no further training of the SKATR backbone, and so a summarized dataset can be constructed. The larger number of resulting patches can be processed by interpolating position encodings, to preserve the total LC size. However, for our HR dataset, the attention operation between $5^3$ times more elements is prohibitively expensive. Further, the physical size of patches in this scheme differs from that used in pre-training, which is likely

suboptimal.

Taking the opposite approach, one can use patches with equivalent physical size to those used in pre-training, but containing more voxels. Now, the computational bottleneck shifts to a new embedding layer, which maps the larger patches into the hidden dimension of the transformer. While a linear layer would introduce too many parameters, 3D-convolutional layers provide a more efficient solution. However, this option uses trainable layers at the input of the network, the transformer backbone must be called at every training iteration, and so there is little efficiency gain.

Alternatively, if the target resolution is known we can pre-train a SKATR network by upsampling LR LCs at each training iteration. The advantage of this approach is that one can select a patch size for the ViTs that suits the target resolution, mitigating computing bottlenecks. The downside is that pre-training must be run with the desired resolution in mind. However, the trained SKATR network can be frozen and used to summarize LCs for a lightweight MLP. Figure 11 shows the result on the HR dataset using this setup. In the majority of parameters, the SKATR-MLP combination has average errors on par with the ViT benchmark, with no clear failure modes. However, the performance for $\zeta$ is not matched by the MLP. Here, a loss of precision is evident, with the spread of predictions around the true value being larger for the MLP.

Finally, a combination of the mentioned solutions might work best. The options that do not repeat SKATR pre-training were not suitable in our example case primarily due to the large gap

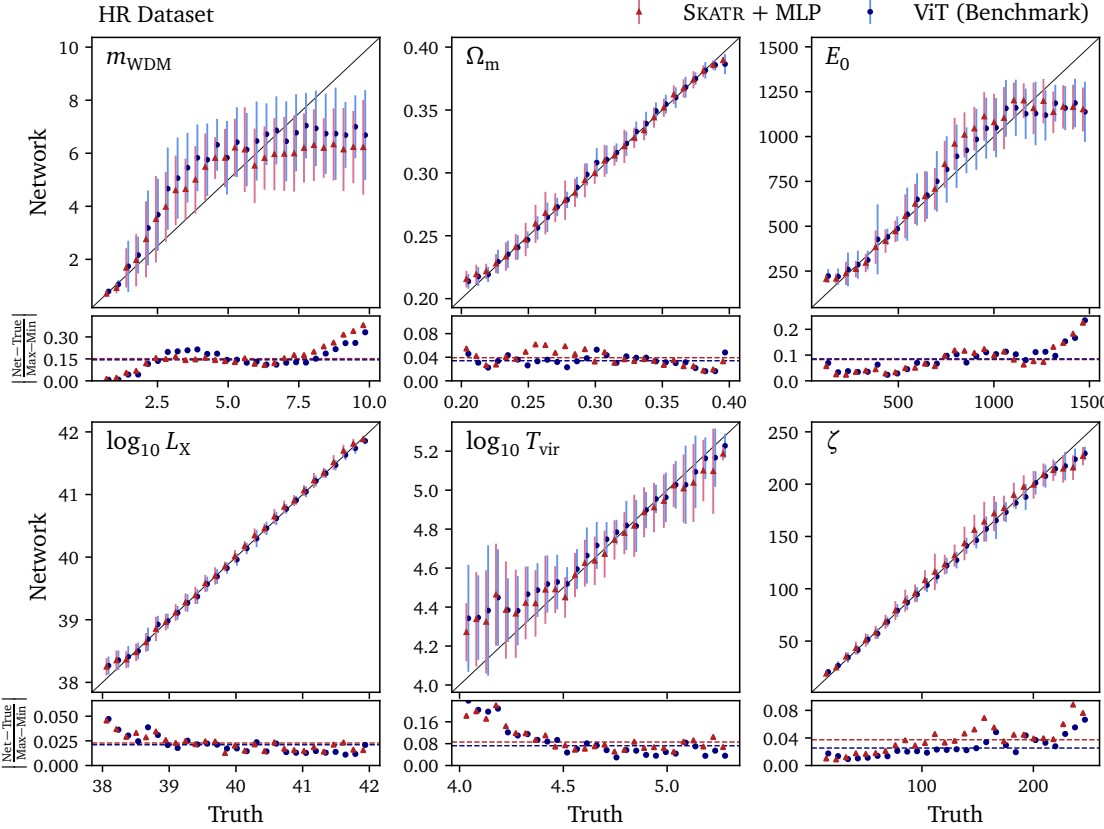

Figure 11: Performance for **SKATR-summarized LCs, trained on the LR dataset, vs ViT trained on the HR dataset**, SKATR is complemented with a 2-layer MLP trained on the HR dataset with LCs upsampled to high resolution at each iteration.

in resolution between LR and HR. A joint approach would be to perform upsampled training at a set of predefined resolutions then use the other methods to bridge any remaining resolution difference.

## 5 Outlook

The complex structure of 21cm images, combined with the impressive data rate expected at the upcoming Square Kilometer Array presents a new challenge to scientific analyses. Machine learning is our only hope to optimally and completely analyze the SKA dataset. However, the size of training datasets is limited by computational expense of high-resolution simulations, as well as memory requirements.

We present SKATR, a vision transformer (ViT) that learns a highly informative summary of 21cm lightcone data using self-supervision. Our analysis showed that SKATR is capable of leveraging large volumes of relatively cheap data to gain performance on high-resolution simulations.

**SKATR finds optimal SKA data representations.** Focusing on regression of astrophysical and cosmological parameters, we first established that ViTs are at least as powerful as the previous CNN benchmark (Figure 4), then demonstrated that SKATR-summarized lightcones contain all information needed to reproduce this performance. In particular, a lightweight MLP trained on frozen SKATR summaries matches the regression accuracy of a full ViT trained from scratch (Figure 5). As a benefit of the SKATR compression, downstream training is extremely cheap, with networks converging over 50 times faster than training from scratch (Figure 6). Also for simulation-based inference, a generative network conditioned on a fixed SKATR summary yields as constraining posteriors as a jointly-trained ViT (Figures 7, 8).

**SKATR generalizes out-of-domain.** Next we showed that the SKATR summary is also maximally informative out of domain, using datasets simulated at high resolution (HR) and low resolution (LR), with and without noise. The combination of frozen SKATR with a small trainable MLP matches the regression performance of a ViT even in the face of novel parameter correlations and observational noise (Figure 9). This is especially important in radio astronomical observations such as with the SKA, where residual systematics remain and always lead to a data-simulation-gap. Summaries obtained through fully-supervised pre-training did not generalise as well as SKATR and performed worse than a ViTs trained from scratch.

**SKATR is data efficient.** When considering regimes with limited training data, we find that SKATR scales more favorably than fully-supervised networks (Figure 10). SKATR therefore represents a promising solution to data constraints related to high-resolution lightcones, namely that they are expensive to simulate and have a large memory footprint.

**Resolution adaptation for SKATR.** Finally, we discussed ways to adapt SKATR to the full resolution dataset. While solutions that customize image patching were not viable in our case due to computational expense, we showed that upsampling the LR data during pre-training produces an informative summary.

A number of interesting directions for future work remain. First, the advantages that SKATR offers in terms of compression can be studied further. At high resolution, even a small dataset of 5k lightcones amounts to almost 1TB of disk space. Large datasets of $O(100k)$ lightcones are therefore unlikely to fit in memory. When training a network from scratch, this will limit the amount of data that can be used. However, SKATR could be trained on some fraction of the data and used to compress all available lightcones, allowing a small network to leverage the entire summarized dataset. It would then be interesting to determine whether the large

dataset retains its statistical power after being summarized by a network that was trained on a small amount of data.

Secondly, our noise model assumed thermal and instrumental noise as well as a foreground avoidance strategy based on the 21cm foreground wedge. Successful generalization for noise models that include further effects such as radio frequency interference and foreground residuals in the EoR window remains to be shown. Given our model successfully transferred between our noised and noiseless scenarios, we expect Skatr representations to remain informative.

Thirdly, the generalization of Skatr representations to new parameters can be further studied in future work, e.g. to lightcones simulated outside of the prior range. An ambitious limiting case is to use a single cosmology for the pre-training, sampling different initial conditions. The question is then whether inference based on this summary remains sensitive to newly varied parameters. Success in this scenario would allow pre-training on SKA observations, providing an implicit bias to combat large uncertainties in simulations.

## Code availability

The code used for this paper can be found at https://github.com/heidelberg-hepml/skatr.

## Acknowledgements

We would like to thank Aaron Nordmann for contributing to this study in the first phase. CH's work is funded by the Volkswagen Foundation. CH also acknowledges funding by the Daimler and Benz Foundation. This work was supported by the by the DFG under grant 396021762 – TRR 257: *Particle Physics Phenomenology after the Higgs Discovery*, and through Germany's Excellence Strategy EXC 2181/1 – 390900948 (the *Heidelberg STRUCTURES Excellence Cluster*). It was additionally supported by the Federal Ministry for Education and Science (BMBF) and the Ministry of Science Baden-Wuerttemberg through Germany's Excellence Strategy.

# A  Lightcone degeneracy in $m_{\text{WDM}}$ and $T_{\text{vir}}$

As discussed in the main text, HR-simulated LCs exhibit a degeneracy in the warm dark matter mass, $m_{\text{WDM}}$, and the minimal virial temperature, $T_{\text{vir}}$. Both parameters bound early star formation and when the limit from $m_{\text{WDM}}$ is stronger than that from $T_{\text{vir}}$, then no information on the latter is available. Here we demonstrate this explicitly using regression and inference results. Figure 12 shows a scatter plot of the predicted and true parameter values by a ViT regressor on the HRDS dataset. Selecting points based on their normalized absolute error (NAE) in $T_{\text{vir}}$ reveals a corresponding cluster in $m_{\text{WDM}}$. In particular, the outliers in $T_{\text{vir}}$ correspond almost directly to the points with low error in $m_{\text{WDM}}$. This suggests that the two parameters are not simultaneously predictable in HRDS lightcones. Figure 13 shows that same result is apparent in the posteriors $q(m_{\text{WDM}}, T_{\text{vir}}|x)$ learned by a cINN + ViT combination. On the left of the figure is a posterior for a test point with $m_{\text{WDM}} > 2.5\,\text{keV}$. Here, the posterior tightly constrains $T_{\text{vir}}$, but has almost maximal uncertainty in $m_{\text{WDM}}$. On the right side of the figure, the converse behavior is observed for an LC with $m_{\text{WDM}} < 2.5\,\text{keV}$.

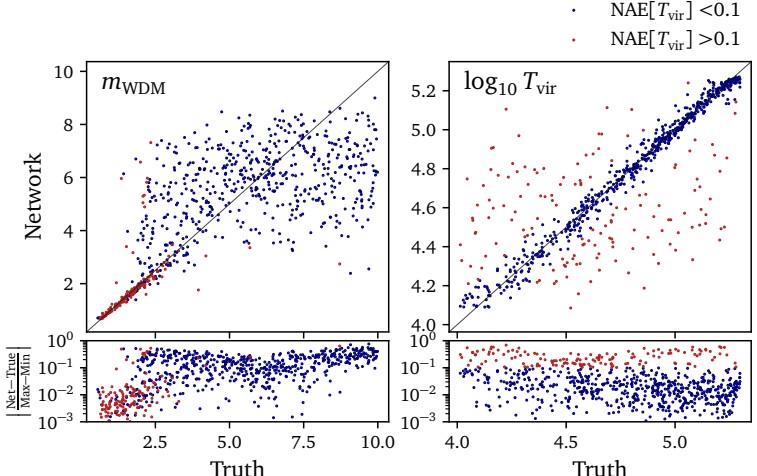

Figure 12: Regression results for a ViT trained from scratch on the HRDS dataset. Points are colored according to absolute error in $T_{\text{vir}}$.

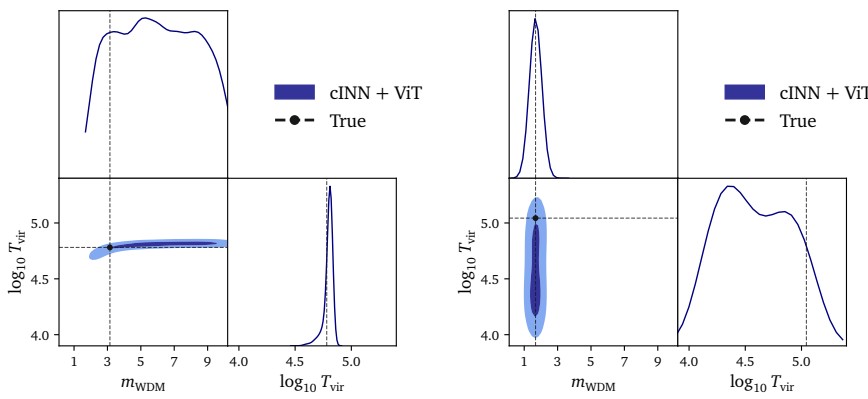

Figure 13: Parameter degeneracy between $m_{\text{WDM}}$ and $T_{\text{vir}}$ reflected in the posterior distributions learned by a cINN + ViT in the HRDS dataset. The test LC in the left plot has $m_{\text{WDM}} > 2.5\,\text{keV}$ and the right plot has $m_{\text{WDM}} < 2.5\,\text{keV}$.

|                     | Encoders               | Predictor     |
| ------------------- | ---------------------- | ------------- |
| Patch size          | (4, 4, 10)             | -             |
| Embedding dim       | 360                    | 48            |
| Attention heads     | 6                      | 4             |
| MLP hidden dim      | 720                    | 96            |
| Blocks              | 6                      | 4             |
| Positional encoding | Learnable sin/cos      | Fixed sin/cos |
| Parameters          | 6.3M                   | 110k          |
| EMA rate $\tau$     | 0.9997                 |               |
| Learning rate schedule | OneCycle            |               |
| Max learning rate   | 0.001                  |               |
| Epochs              | 1000                   |               |
| Batch size          | 64                     |               |
| Optimizer           | AdamW                  |               |
| Weight decay        | 0.001                  |               |

Table 2: Network and optimization hyperparameters used in SKATR pre-training, discussed in Section 3.

## B   Further training details and hyperparameters

**Pre-training**

A key component of the SKATR pre-training loss in Eq. (8) is the mask sampling procedure, appearing as $p_{\text{mask}}(M)$. For this sampling, we follow the strategy outlined for video in Ref. [37]. This involves a combination of "long-range" and "short-range" masks, which both span the full redshift (time) dimension of the LC, but have different spatial structure. Long range masks are constructed by sampling three rectangles, each with an aspect ratio in the range [0.75, 1.5] and 70% coverage of the spatial area, then taking their union. Short range masks are sampled similarly, but using eight rectangles with 15% coverage. Again, the masks extend across the entire redshift axis.

In the predictor network $h_\psi$ it is not necessary to apply a patching step. This is because its input is already a set of embeddings—those of the context patches. In order to make predictions at the target locations, a set of "mask tokens" are added to the input set. These mask tokens are constructed by summing a shared learnable vector with the positional encoding for each target patch. The full set of patches (context and mask) are then embedded into the hidden dimension of the predictor using a shared linear layer. Similarly, at the predictor output another linear layer projects into the embedding dimension of the target patches.

In order to improve efficiency in training, multiple masks can be sampled per LC. The loss is then calculated for each mask and averaged before taking a gradient step. This saves one evaluation of the context encoder per additional mask. In SKATR, we sample two long-range and two short-range masks per LC.

In Table 2, we give the full list of hyperparameters for the SKATR networks presented in this paper. With a single NVIDIA H100 GPU, pre-training takes about 50 hours and uses 45GB of GPU memory at batch size 64.

|                          | LR/HRDS        | HR           |
|--------------------------|----------------|--------------|
| Patch size               | (4, 4, 10)     | (7, 7, 50)   |
| Embedding dim            | 144            | 96           |
| MLP hidden dim           | 288            | 192          |
| Attention heads          | 4              |              |
| Blocks                   | 4              |              |
| Positional encoding      | Learnable sin/cos |           |
| Patch aggregation        | Mean           |              |
| Head network             | MLP            |              |
| Parameters               | 690k           | 540k         |
| Loss                     | Mean Absolute Error |         |
| Learning rate schedule   | Constant       |              |
| Learning rate            | $10^{-4}$      | $3 \cdot 10^{-4}$ |
| Epochs                   | 1000           |              |
| Patience                 | 50             |              |
| Batch size               | 32             |              |
| Optimizer                | AdamW          |              |
| Weight decay             | $10^{-3}$      |              |

Table 3: Network and optimization hyperparameters used to train ViTs for regression. The MLP head architecture is given in Eq. (3).

**Regression**

Table 3 lists the hyperparameters we use when training ViTs for regression, including when pre-training for the result in Figure 10. For MLPs trained on top of pre-trained summary networks, a faster learning rate of $5 \cdot 10^{-4}$ is used. Aside from this, the optimization hyperparameters are shared with the ViT. Similarly, the CNN in Figure 4 is trained using the same optimization settings, but a learning rate of $3 \cdot 10^{-4}$. Its architecture matches exactly the description in Ref. [46]. Training times for regression were measured using a single NVIDIA A30 GPU.

Note that the ViT architecture used in SKATR pre-training is larger than the ViTs trained from scratch, with two extra blocks and a wider embedding dimension. We found that training from scratch with the larger network resulted in overfitting and thereby reduced performance. Using a smaller ViT for SKATR degraded performance slightly, but does not strongly affect the results. We understand the lack of overfitting when pre-training to be a consequence of the fact that the loss is based on masking. This means that a single lightcone can provide multiple distinct training objectives, effectively increasing the data efficiency.

**Inference**

Table 4 lists the hyperparameters we use when training cINNs for posterior estimation. ViTs trained as summary networks share the same architecture as for regression (Table 3), except that no head network is used.

## C    Additional plots

Here we present a selection of supplementary plots:

- Training times for downstream regression on the HRDS dataset [Figure 14]. Similarly to the timing result in the main text, we see more than 50× speed-up in convergence for the MLP

| | |
|---|---|
| Bijector | Rational quadratic spline |
| Spline bound | [-10, 10] |
| Spline bins | 10 |
| Block type | Coupling |
| Blocks | 6 |
| Layers per block | 2 |
| Layer dim | 128 |
| Channel mixing | Fixed rotation |
| Latent distribution | Unit Gaussian |
| Parameters | 350k |
| Learning rate schedule | Constant |
| Learning rate | $10^{-4}$ |
| Epochs | 1200 |
| Patience | 100 |
| Batch size | 64 |
| Optimizer | AdamW |
| Weight decay | $10^{-3}$ |

Table 4: Network and optimization hyperparameters used to train cINNs for posterior estimation.

compared to training a ViT from scratch. The summarization time is also less significant in this dataset.

- Posterior likelihoods and calibration:
  - CNN vs ViT on HR dataset [Figure 15]. The ViT summary network yields higher posterior likelihoods than the CNN on average. Both networks are well calibrated.
  - ViT vs SKATR with and without XAttn pooling on HRDS dataset [Figure 16]. Using the XAttn pooling in SKATR slightly improves the average posterior likelihood, though both approaches are very slightly outperformed by the ViT trained from scratch. The calibration curves are all equally overconfident.
- 1D posteriors
  - CNN vs ViT on HR dataset [Figure 17]. The ViT summary network improves over the CNN in all parameters, with the largest difference in $E_0$.
  - ViT vs XAttn[SKATR] on HRDS dataset [Figure 18]. Using an XAttn pooling to aggregate SKATR summary patches gives marginal posteriors that are equally constraining as the ViT trained from scratch.
- 2D posteriors
  - ViT vs SKATR on LR dataset [Figure 19]. The posteriors from both methods largely agree.
  - ViT vs XAttn[SKATR] on HRDS dataset [Figure 20]. The posterior from XAttn[SKATR] constrains $m_{\mathrm{WDM}}$ much more tightly than the ViT, but is slightly wider in the other parameters.

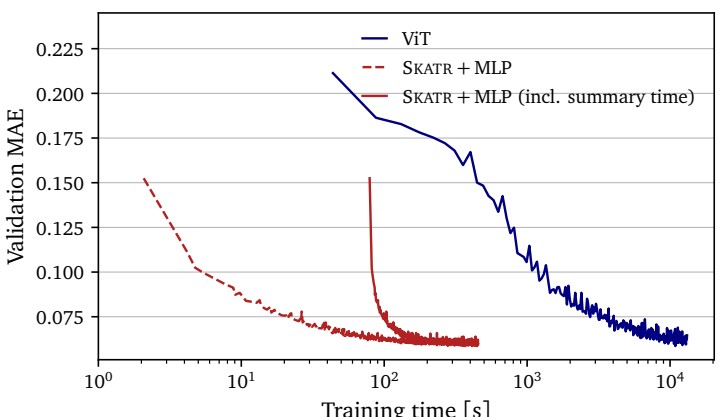

Figure 14: Same as Figure 6, but for downstream training on the HRDS dataset.

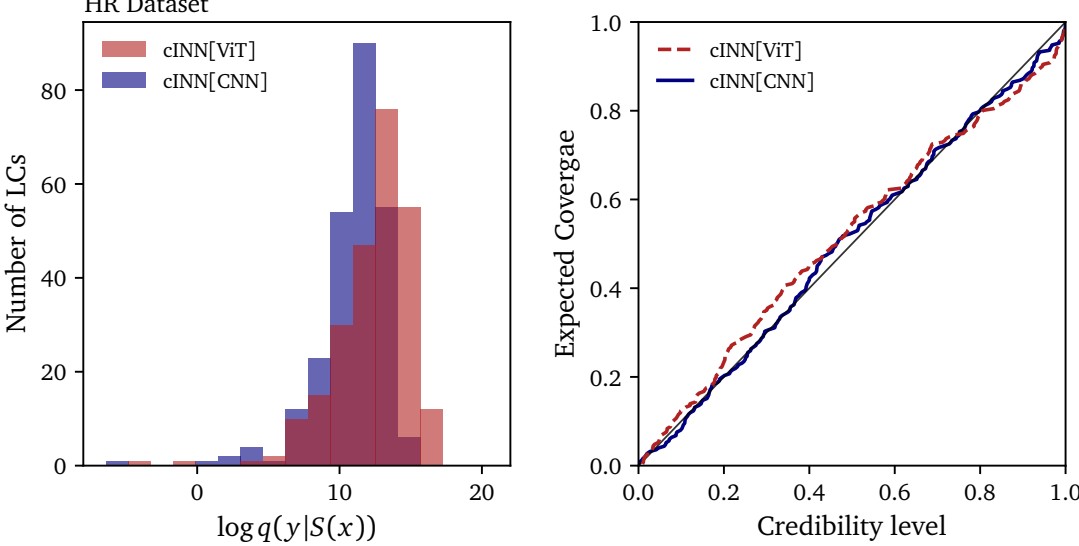

Figure 15: Same as Figure 7, but comparing CNN and ViT summaries on the HR dataset.

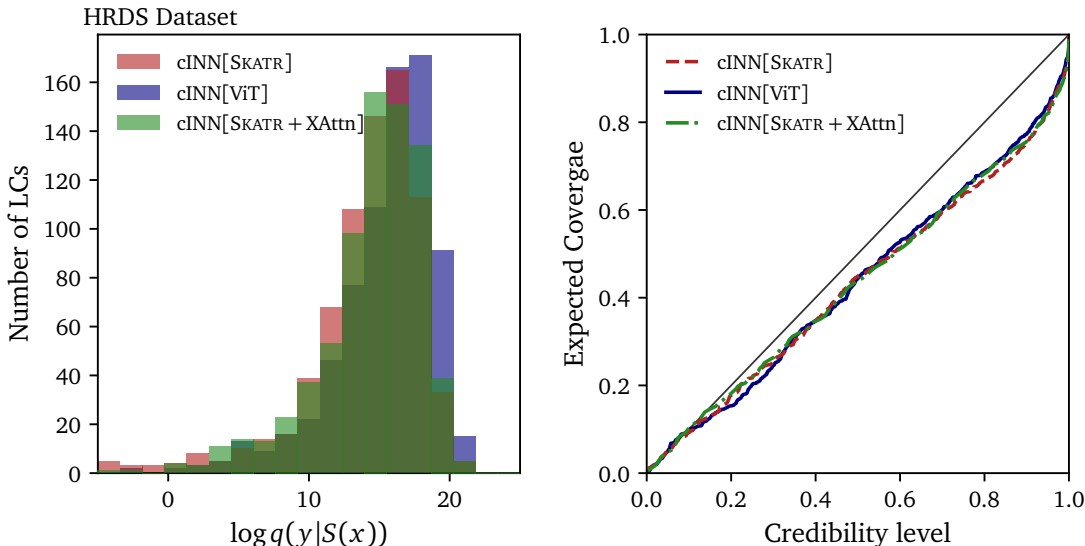

Figure 16: Same as Figure 7, but for the HRDS dataset and including a trainable XAttn layer over frozen SKATR embeddings.

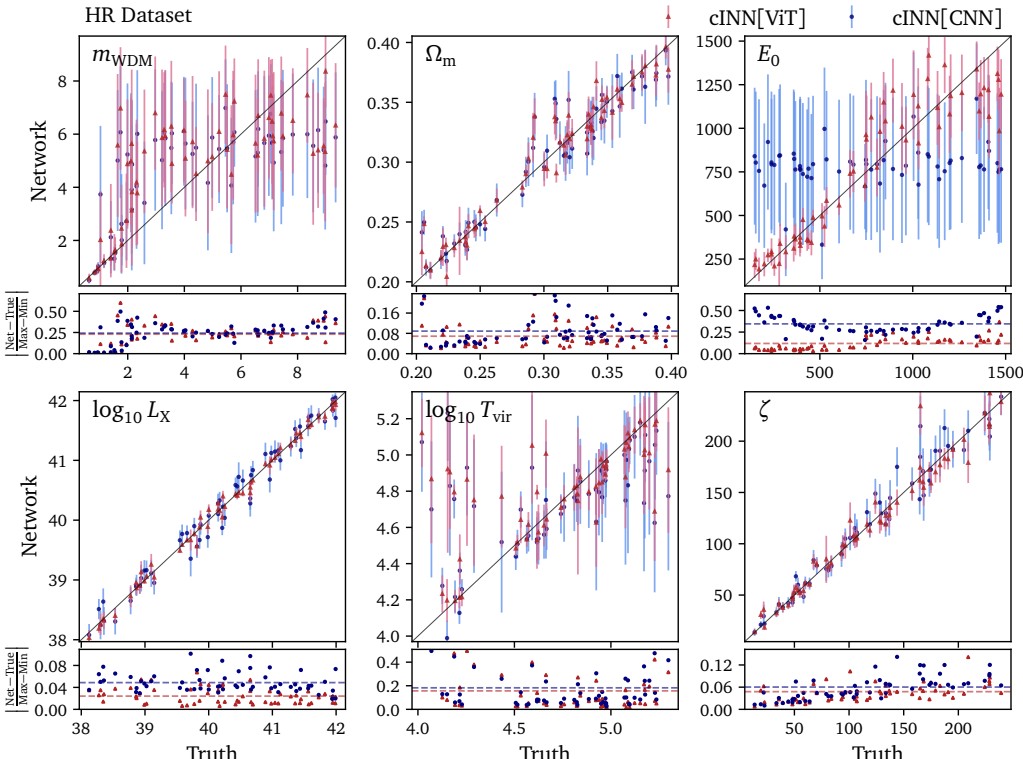

Figure 17: Same as Figure 8, but comparing CNN and ViT summaries on the HR dataset.

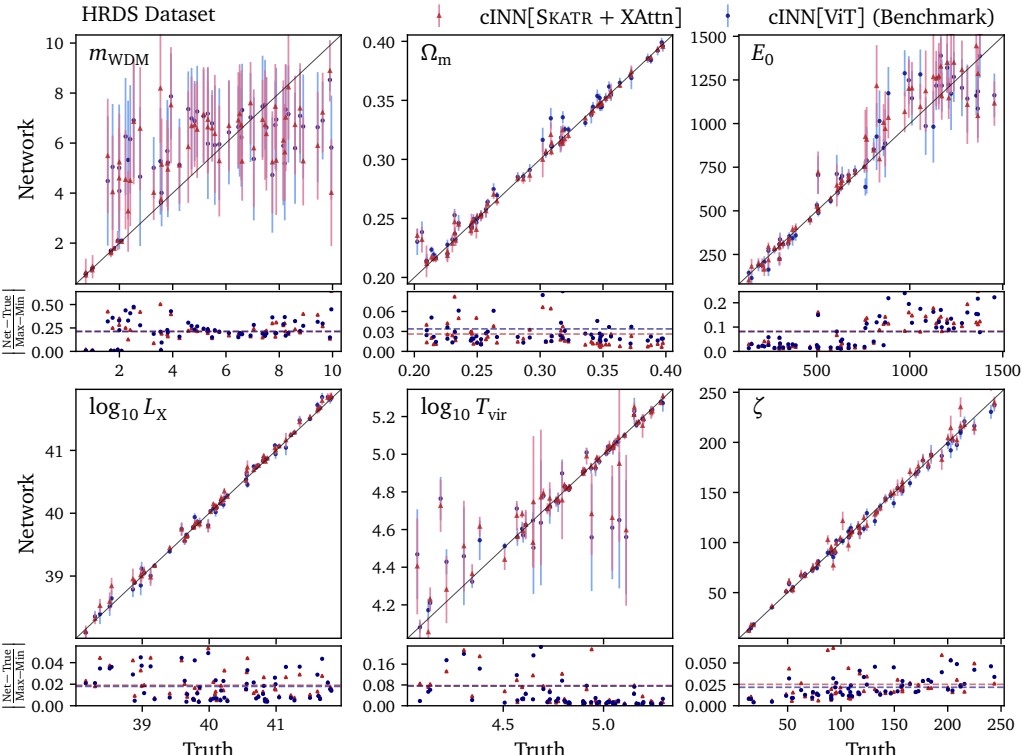

Figure 18: Same as Figure 8 but for the HRDS dataset and including a trainable XAttn layer over frozen SKATR embeddings.

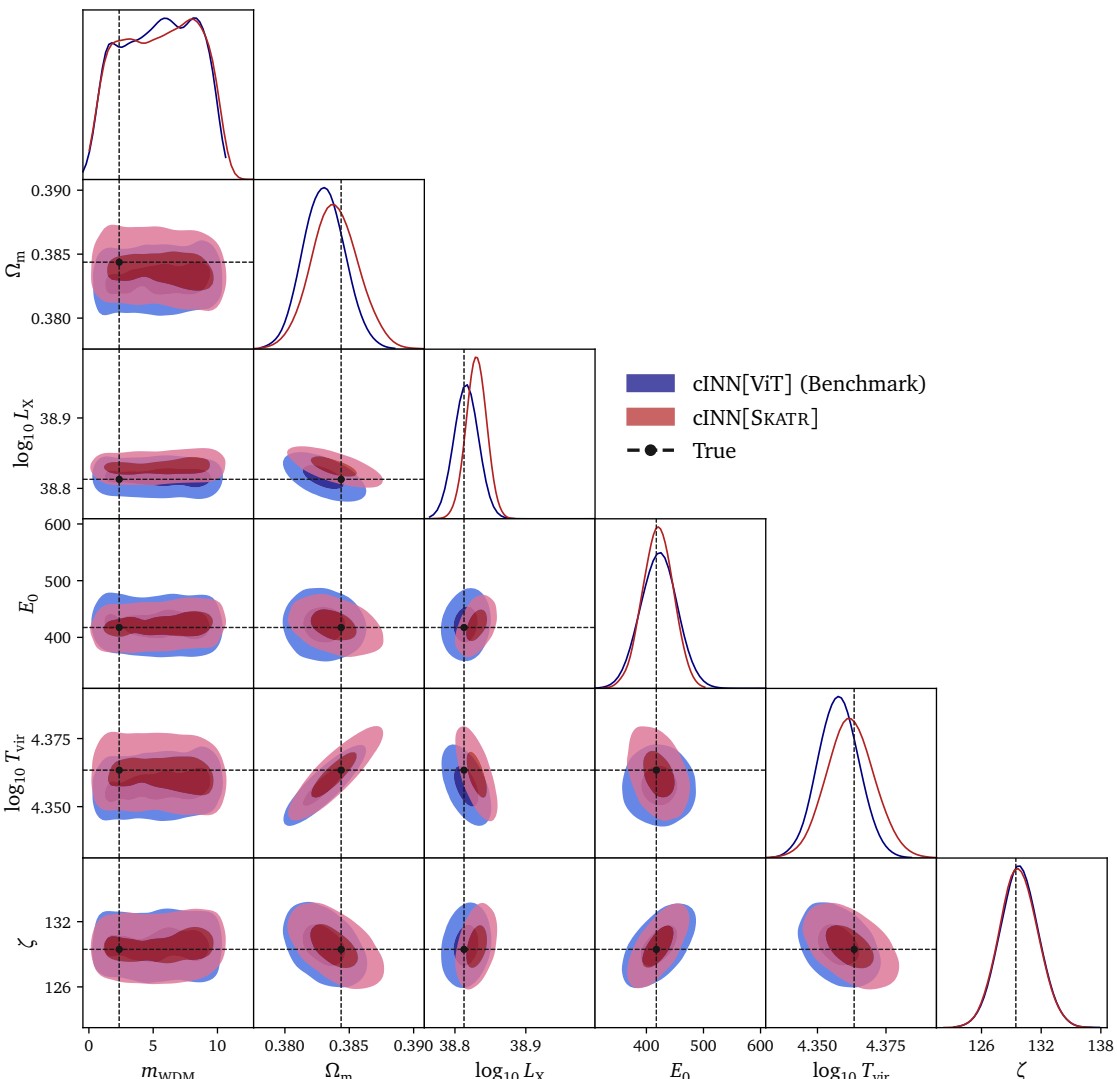

Figure 19: Comparison of posterior distributions sampled from a cINN trained jointly on HRDS LCs summarized by either: A complete ViT (blue) or frozen SKATR network (red). The shading levels indicate the 68% and 95% highest-density credibility regions.

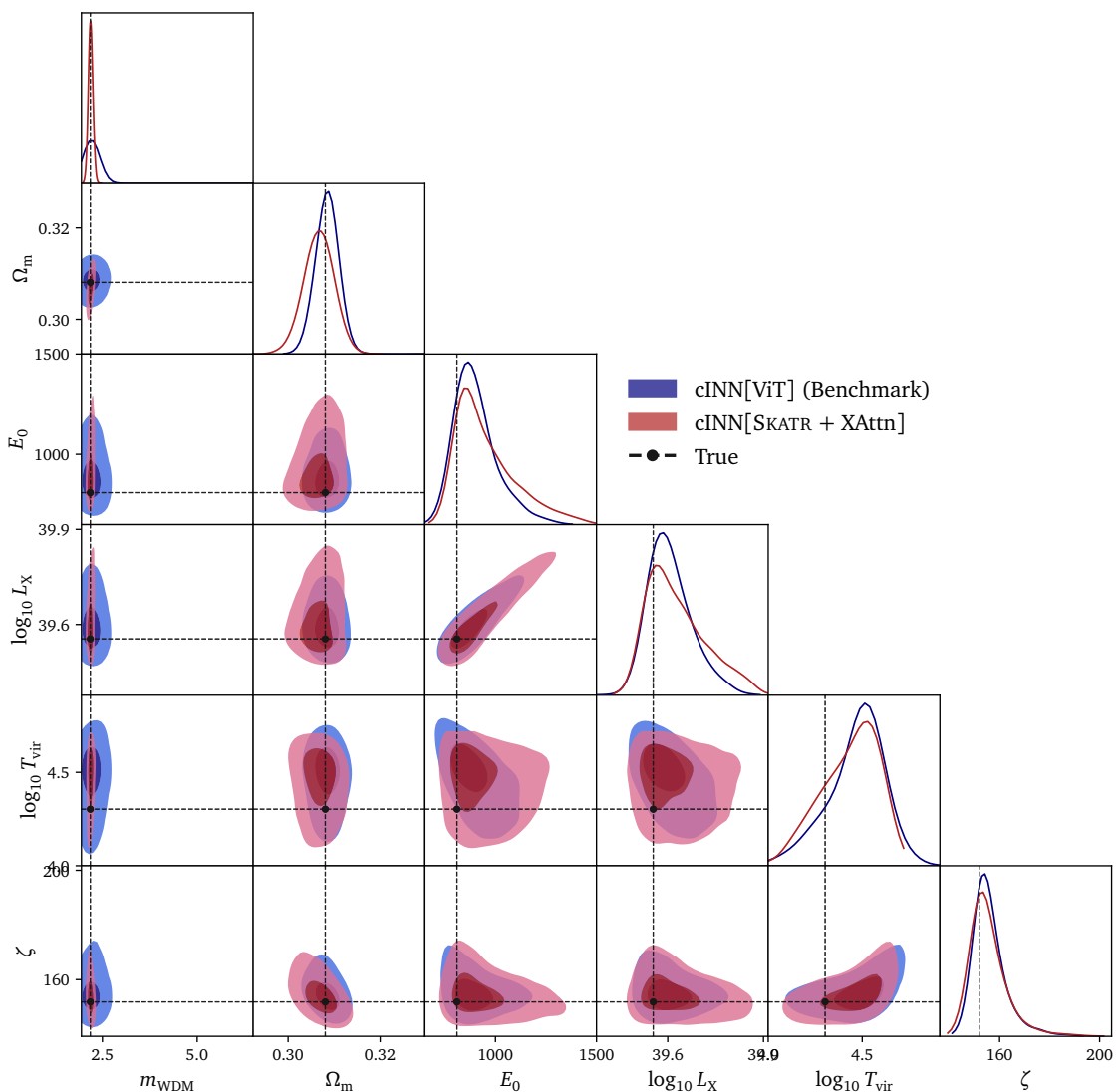

Figure 20: Comparison of posterior distributions sampled from a cINN trained jointly on HRDS LCs with either: A complete ViT (blue) or a XAttn pooling over frozen SKATR embeddings (red). The shading levels indicate the 68% and 95% highest-density credibility regions.

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
