# Peer review of "SKATR: A Self-Supervised Summary Transformer for SKA"

_SciPost Physics, doi:SciPost Phys. 18, 155 (2025)_

## Round 1 · Referee Report · Anonymous (Referee 1) · 2025-4-15

Report

I believe this satisfies the journal criteria, and I recommend its publication.

Recommendation

Publish (easily meets expectations and criteria for this Journal; among top 50%)

---

## Round 1 · Referee Report · Anonymous (Referee 2) · 2025-4-15

Strengths

The authors provide an interesting vision transformer (ViT) application that has not yet been applied to an SKA-related project. The authors successfully demonstrate the advantages compared to consolidated machine learning applications as well as the challenges due to the degeneracy of the 21-cm astrophysical parameters.

Report

The paper meets the journal's criteria. The author addressed the remarks and corrections pointed out in the previous review. I recommend its publication.

Requested changes

No additional changes required.

Recommendation

Publish (easily meets expectations and criteria for this Journal; among top 50%)

---

## Round 1 · Author Response

We thank the referees for their thoughtful comments and suggestions. We have accordingly updated the manuscript with the requested changes, listed below.

---

## Round 1 · List of Changes

Response to report #1

  1. In Sec 2.1, the second sentence: "... leading to LCs with 140 voxels in the on-sky axes ..." can be more explicit in specifying that 140 is the dimension along each on-sky axes. Like "... leading to LCs with 140 voxels along each of the two on-sky axes ...". Following the suggestion, we reworded to “along each on-sky axis”.

  2. In Sec 2.1, why are the LR datasets downsampled by a factor of 2.5? It is unclear to me if this is to reduce compute demand or perhaps to average out some of the noise. If so, why isn't the HR dataset also downsampled by the same factor? If this is a misunderstanding on my part, please let me know. Otherwise, I would advise explicitly stating reasons in the section. The reason we downsample the LR simulations is indeed to reduce computational cost — producing all the results of this paper at the original size (70, 70, 1175) would take significantly more resources. Then, in order to enable a fair comparison between networks trained on HR and LR, we want to equate the image sizes. For this reason, the 2.5x downsampling of the LR light cones is chosen to match a 5x downsampling of the HR dataset (HRDS). We reworded the description of these datasets in an attempt to make this clearer.

  3. Sec 2.1, last paragraph is missing the specification for HRDS's train-test-validation datasets. We have fixed this, indicating that HRDS and HR have the same splits.

  4. Sec 3.1, last para. Given the statement: "In order to manage the computational cost of a ViT, the patch size should be selected with an expected image resolution in mind", can the authors provide a reasoning behind the specific choice of patch sizes (7, 7, 50) for HR and (4, 4, 10) for LR/HRDS, considering the latter has a constant downsampling factor of 5 about each dimension? If not from a rigorous analysis, at least an empirical reasoning. The patch sizes are chosen by hand, with the primary constraint being that they should exactly divide the lightcone. We then approximately maximise the total number of patches (with similar number in each axis) while remaining within memory limits. We did not perform a systematic hyperparameter scan and, as such, the values are not necessarily optimal. We have added a clarification along these lines to the relevant section. (Now paragraph 3 in Sec 3.1)

  5. Figure 6, right: Have the authors explored a post-hoc or post-training calibration (such as Bayesian posterior refinement) of SKATR/ViT to account for the consistently conservative posterior estimates? While we certainly agree that such a procedure should be included in a complete inference study, we consider it to be beyond the scope of this paper. Here we are satisfied with showing that there are no significant differences between the approaches in terms of calibration. However, neither of our methods introduce any technical limitations that would prohibit post-hoc calibration.

  6. Unless I have missed this in the text, could the authors please specify the computational facilities (RAM/number/type of GPUs or a rough estimate of total compute) used in the training of SKATR and ViT? - Especially useful so one may better interpret how the training times quoted in Figures 5 & 14 scale with one's native implementation of SKATR, ViT. We added the hardware information (and timing for SKATR pre-training) to Appendix B Note that the times reported in figures 5 & 14 do not scale with the SKATR training time, since the summary network is frozen for regression.

  7. Appendix C: For the benefit of the reader, could the authors please elaborate on the implication of the bullet points? A short discussion of each figure (Figures 14-20) would suffice. We have expanded each of the points with a brief explanation.

  8. Could the authors please further elaborate on plans for the SKATR with regard to areas of improvement and/or application? We have added discussion of possible future developments of SKATR. These are: (a) Further study to quantify benefits compression (b) Further study on sensitivity to unseen parameters

Response to report #2

  1. Connected to the question 3, here below. If galactic and/or extra-galactic foreground contamination is not included in this paper, I suggest adding a few sentences on page 2 after the sentence that ends with "...absent during the pre-training and with noised data" mentioning its absence and stating that, residual foreground contamination (due to the imperfect subtraction) could make the application of ML application more challenging. Besides instrumental and thermal noise, we follow what is also called the foreground avoidance strategy (as opposed to actively modelled foregrounds); for the 21cm signal the foreground contamination is largely restricted to the foreground wedge which dominates at low k_parallel (Morales et al. 2012; Liu et al. 2014). We assume a foreground scenario where the 21cm foreground wedge in k-space extends to the primary field-of-view of the instrument (SKA), following the 'optimistic' setting for the 21cmSense code (Pober et al. 2013, 2014). We explicitly zero out these modes in our ‘Noised’ lightcones. The so-called EoR window outside the wedge is assumed to be clean of residual foregrounds and systematics. Foreground avoidance is also the strategy for some of the SKA percursors. We have updated the ‘Noised LCs’ paragraph in section 2.1 to state this explicitly.

  2. In the title of section 2 "Lightcones" (delete space) We have updated this here and throughout the text.

  3. In section 2.1, page 4, the sentence: "The thermal foreground noise estimate considers the 21cm foreground wedge in k-space to cover the primary field-of-view of the instrument" is not clear if the author is including galactic and/or extra-galactic foreground contamination in the analysis or only systematic noise. I suggest rephrasing the sentence and making it clearer as the text appears to suggest that foreground contamination is included when my understanding is that is not. The foreground wedge covers of a combination of foregrounds and instrument systematics (mode couplings) that all occupy the wedge region for interferometers such as the SKA. This includes dominant 21cm foregrounds such as Galactric synchrotron and extragalactic point sources (where signals spectrally smooth are mapped to ~low k_parallel). We have now specified this in the ‘Noised LCs’ paragraph.

  4. Page 5, in equation (3). I am a bit confused by the notation. The input of equations 3 and 4 shouldn't be the union of the context encoder output: i.e. \tilde{z} and the prediction: i.e. p, so embedding: p \cup \tilde{z}, rather than the input x? Indeed, in equations 3 and 4, the functions being defined act on the set of patch embeddings output by a ViT. The inputs are not x as defined w.r.t equation 5. We updated the formulas to use $z$ for the input, in accordance with the following section on pretraining. We hope this is now clear in combination with our other changes in response to your point 7 below.

  5. Page 5, sentence after equation 3. Is not clear to me what the W1 and W2 are. Are these the weights of the two-layer dense network? also, It would be helpful to remind the reader that dimension 6 is the dimension of the astrophysical parameters. W1 and W2 are indeed learnable weights of the dense network. We clarified the wording around this equation as suggested, and added that 6 is the number of parameters.

  6. Page 6, sentence after equation 4. Similar question as for equation 3. Are Wk and Wv the weights of this second network? Also, it is not clear the dimension of the learnable parameter: q. Yes, Wk and Wv are also learnable weights of the ‘XAttn’ layer. The parameter q is a dx1 matrix: the “single token with dimension d”. We updated the wording of this sentence to improve the clarity.

  7. Page 6, paragraph starting with "In a ViT, ..." Is not clear to me when and how the two task-specific networks are employed. From the paragraph, the author is using a combination of the two-layer dense network (Eq 3) and a dynamic pooling function (Eq 4), but in the result later only the MLP is mentioned. Moreover, the author mentions that in the pre-training step, the JEPA loss is employed directly in the embedding. Meanwhile, the result related to Figure 3, appears that the MLP is employed to predict the 6 parameters. It would help the reader to include a small paragraph that clarifies when the two aggregation steps are used in the pipeline (during the pre-training or validation, etc.), their task (my understanding is for results in Fig 3), at which stage they have been trained, etc. We attempted to improve the clarity by: (a) Exchanging the order of the paragraphs starting “In a ViT…” and “Depending on the task…” (b) Rewording to highlight when each option is used (c) Relocating the summary figure from Section 6. as suggested in your point 13. Also note that the final paragraph of section 3 explains that after pre-training, the SKATR summary is always defined as the mean over patch embeddings.

  8. Page 6, at the very beginning of section 3.2, "Lightcones" (delete space) Addressed in point 2.

  9. Page 6, sentence: "During training, an LC (batch) is divided into a set of N patches...". Is this N the same dimension as shown in Figure 1 "Transformer Blocks (xN)"? If not please change to avoid confusion. Indeed, these are different numbers. We revised the notation for the number of patches to $n$.

  10. Page 7, sentence: "Finally, a smaller transformer...". Define what you mean by smaller, are you using the same architecture as the context and target encoder but with different embedding dimensions? The embedding dimension is indeed smaller, which we now specified in the text. The full specifications of the network are in Appendix B, Table 2.

  11. Page 7, sentence after equation 9. Maybe I missed it, but what value has been used for tau? The value is given in Appendix B, Table 2: We use 0.9997.

  12. Page 13, sentence: "The only exceptions are LX , where the improvement is only marginal, and E0, where SKATR is slightly outperformed". Did the author test if this trend is present also in the LR dataset? if available, it would be interesting to mention in just one sentence. Since the LR dataset is the one used for pretraining, it is not informative to track the scaling with data as we have in Figure 9 — the same volume of data would be available for pretraining and finetuning. In addition, training with a regression-based summary network would be essentially equivalent to training from scratch in this case. As such the relevant plots are Figure 4: The SKATR compression yields equivalent regression performance to the full lightcone.

  13. Page 16, I suggest moving Figure 11 to Section 3. This Figure helps to understand what is explained in part 3.1 as it would help to answer my question 7. We have moved the plot to Section 3 as suggested.

  14. in section 5: "Outlook". For completeness, there should be a few sentences regarding the absence of residual foreground contamination and other observational systematics, e.g. residual foreground contamination, beam effect, RFI corrupted data, etc. Although I understand this work is focusing on testing the ML application, it is important to remind the readers that the presence of contamination in future 21-cm SKA observations can impact the efficiency of these predictions and remind once again of the importance of taking this into account in future applications to actual observational data. We added a paragraph to this effect in the outlook.

---

## Editorial Decision

published